# A New Target of Dental Pulp-Derived Stem Cell-Based Therapy on Recipient Bone Marrow Niche in Systemic Lupus Erythematosus

**DOI:** 10.3390/ijms23073479

**Published:** 2022-03-23

**Authors:** Soichiro Sonoda, Takayoshi Yamaza

**Affiliations:** Department of Molecular Cell Biology and Oral Anatomy, Kyushu University Graduate School of Dental Science, Fukuoka 812-8582, Japan; ilikeanimalaso@dent.kyushu-u.ac.jp

**Keywords:** dental pulp, mesenchymal stem cells, systemic lupus erythematosus, cellular microenvironment, immunomodulation, cell-cell interaction

## Abstract

Recent advances in mesenchymal stem/stromal cell (MSC) research have led us to consider the feasibility of MSC-based therapy for various diseases. Human dental pulp-derived MSCs (hDPSCs) have been identified in the dental pulp tissue of deciduous and permanent teeth, and they exhibit properties with self-renewal and in vitro multipotency. Interestingly, hDPSCs exhibit superior immunosuppressive functions toward immune cells, especially T lymphocytes, both in vitro and in vivo. Recently, hDPSCs have been shown to have potent immunomodulatory functions in treating systemic lupus erythematosus (SLE) in the SLE MRL/*lpr* mouse model. However, the mechanisms underlying the immunosuppressive efficacy of hDPSCs remain unknown. This review aims to introduce a new target of hDPSC-based therapy on the recipient niche function in SLE.

## 1. Introduction

We systematically searched PubMed and ISI web of knowledge databases for eligible articles involving dental pulp, mesenchymal stem/stromal cells, immunomodulation, and/or systemic lupus erythematosus. The current treatment for patients with autoimmune diseases, such as systemic lupus erythematosus (SLE), rheumatoid arthritis, type I diabetes, and multiple sclerosis, is pharmacotherapy with chemical and biological drugs, including steroids and monoclonal antibodies. The immunomodulatory efficacy of mesenchymal stem/stromal cells (MSCs) has been shown in a large number of preclinical and clinical studies investigating cell therapies [1]. Currently, MSC therapy for graft-versus-host disease and Crohn’s disease is only approved for use in Canada, New Zealand, Japan, South Korea, and Europe (France and Italy) due to a lack of consistent efficacy data [2,3]. Human dental pulp-derived MSCs (hDPSCs) have recently been identified in the dental pulp tissue of teeth. hDPSCs are considered a novel alternative to treat SLE because of their potent immunosuppressive function in vitro. However, the mechanisms underlying the immunosuppressive efficacy of hDPSCs remain unknown. This review provides a new target of hDPSC-based therapy for SLE, primarily through hDPSC-released extracellular vesicles (EVs).

## 2. Characterization of hDPSCs

Adherent colony-forming fibroblast-like spindle-shaped cells were identified in the bone marrow stromal cells as colony-forming unit fibroblasts (CFU-Fs), which are different from bone marrow hematopoietic stem cells [4]. Non-HSC bone marrow stromal cells exhibit self-renewal and multipotency into various mesenchymal cell lineages, such as osteoblasts, adipocytes, and chondrocytes; the term MSCs was proposed and remains the most frequently used term [5]. MSCs have been found in various mesenchymal tissues and organs, such as the adipose tissue, liver, muscle, and umbilical cord [6]. Recently, the International Society for Cellular Therapy (ISCT) proposed the following minimum criteria for characterizing MSCs [7]: MSCs (1) exhibit adherence on a plastic culture dish/flask; (2) are positive for CD105, CD90, and CD73, but negative for CD34, CD45, CD14, CD11b, CD79 alpha, or CD19, and human leukocyte antigen DR (HLA-DR); and (3) show multi-differentiation capacity into osteoblasts, adipocytes, and chondrocytes in vitro.

Among several hard tissues in the human body, dentin is the only hard tissue whose formation is lifelong under physiological conditions, even when under pathological conditions, namely, secondary and tertiary dentin (reparative dentin), respectively [8,9,10]. Therefore, dental pulp tissue has been considered to contain MSCs. In the early 2000s, MSCs were identified in the dental pulp tissue of permanent teeth and referred to as dental pulp stem cells (DPSCs) [11]. MSCs were identified in the remnant dental pulp tissue of exfoliated human deciduous teeth and apical papilla of developing tooth roots and referred to as stem cells from exfoliated human deciduous teeth (SHED) [12] and stem cells from apical papilla (SCAP) [13], respectively (Figure 1). MSCs were subsequently found in the dental pulp tissue of supernumerary teeth and are referred to as human supernumerary teeth-derived stem cells (SNTSCs) [14] (Figure 1). These hDPSC subpopulations commonly possess clonogenic properties with self-renewal and in vitro multipotent capability into odontoblasts/osteoblasts, adipocytes, and chondrocytes. They exhibit adherent CFU-F-forming capacity on a plastic culture dish/flask and express a typical MSC immunophenotype positive for CD146, CD105, CD73, and CD90, and negative for CD34, CD45, CD14, CD11b, CD19, and HLA-DR (Figure 1). They also have a high population doubling rate and low telomerase activity. hDPSCs have an in vivo tissue regeneration capability to form dentin/pulp and bone [15,16]. Therefore, hDPSCs are an ideal source for the regeneration of oral and maxillofacial bones and dental pulp [17,18]. 

Recently, we have focused on stem cell potency of regenerative medicine in hDPSCs. hDPSCs showed potent immunosuppressive function [14,15,19,20] and exhibited the unique ability of transdifferentiation into hepatocytes and cholangiocytes [21,22,23]. The cell aggregation technique can be used to generate hDPSC-based bone and liver microtissues [16,24,25]. Recently, disease-specific hDPSCs have been characterized for their pharmacological rejuvenation [10,26,27,28,29,30]. Thus, hDPSC-based therapy is considered a novel option in regenerative medicine [31]. Indeed, the clinical application of hDPSCs has started in dental pulp regeneration of injured teeth [32]. Recently, quality-controlled clinical-grade hDPSCs were manufactured for regenerative medicine applications [33,34].

## 3. Intercellular Interaction between hDPSCs and T Lymphocytes

Three intercellular mechanisms play a role in the immunomodulatory effects of hDPSCs on the proliferation, differentiation, and apoptosis of T lymphocytes: (1) direct cell–cell contact of hDPSCs to induce cell death of T lymphocytes; (2) indirect action of hDPSCs through paracrine factors and EVs to T lymphocytes; and (3) cell death (apoptosis) of hDPSCs caused by T lymphocytes and efferocytosis of the apoptotic bodies caused by macrophages (Figure 1). When hDPSCs are co-cultured with T lymphocytes activated with plate bounded anti-CD3 and soluble anti-CD28 antibodies, hDPSCs induce the differentiation of CD4^+^CD25^+^Foxp3^+^ regulatory T cells (Tregs) and the release of interleukin 10 (IL-10), following stimulation with transforming growth factor-beta (TGFB)/IL-2. hDPSCs also suppress the differentiation of CD4^+^IL^-^17^+^ interferon-gamma^−^ (IFNG^−^) T helper 17 (Th17) cells and IL-17 release upon stimulation with TGFB/IL-6 [15,19,35]. Interestingly, hDPSCs express both Fas and FasL [36]. FasL-expressing hDPSCs can induce apoptosis of Fas-expressing T lymphocytes and suppress the activated T lymphocyte proliferation. In contrast, an in vitro study demonstrated that FasL-expressing activated T lymphocytes induce apoptosis in Fas-expressing hDPSCs [15]. Apoptotic hDPSCs show immunosuppressive efficacy in vivo [37], suggesting that the apoptotic phenomena of donor hDPSCs and recipient T lymphocytes contribute to this efficacy. hDPSCs can also release multiple paracrine factors to regulate the proliferation and differentiation of T lymphocytes [38]. We summarized the molecules involved in the immunosuppressive function of hDPSCs (Table 1). hDPSCs could potentially modulate the proliferation, function, and death of T lymphocytes. Recently, due to their potent immunomodulatory properties, hDPSCs have been investigated for the treatment of SLE, experimental colitis, and asthma [15,35,36].

## 4. Therapeutic Effects of hDPSCs in the SLE MRL/*lpr* Mouse Model

SLE is characterized by disorders such as the production of autoantibodies and the deposition of immune complexes in multiple organs. Immune tolerance is critically eliminated by various self-antigens in SLE because of the dysregulation of the innate and adaptive immune systems. Among the several types of CD4^+^ T lymphocytes, abundant IL-17-releasing Th17 cells play a significant role in the pathogenesis of autoimmune diseases [56]. In contrast, Tregs suppress helper T lymphocytes to prevent abnormal immune responses to self-antigens, resulting in persistent tolerance [57]. SLE is associated with an imbalance in the Tregs to Th17 cell (Treg/Th17) ratio [58]. MRL/*lpr* mice express the mutant *lpr* in the gene encoding Fas protein and spontaneously develop many lupus autoantibodies and manifestations, such as lymphadenopathy, arthritis, and nephrosis [59]. MRL/*lpr* mice typically express increased serum levels of autoantibodies, including anti-double-strand DNA (dsDNA) IgG, anti-dsDNA IgM, and anti-nuclear antibodies (ANAs). They also develop refractory renal disorders, such as nephritis with glomerular basal membrane disorder, mesangial proliferation, proteinuria, and elevated serum creatinine levels. These symptoms are also observed in patients with SLE, indicating that MRL/*lpr* mice represent a viable animal model for SLE. Recently, systemic transplantation of hDPSCs (1 × 10^5^ cells per 10 g body weight) has demonstrated adequate therapeutic efficacy for SLE disorders in MRL/*lpr* mice (female, 16-week-old) [15,19,36]. Four weeks after systemic transplantation, hDPSC transplantation significantly reduced the serum levels of autoantibodies, including anti-dsDNA and ANA antibodies. In addition, hDPSC transplantation further improved renal disorders such as nephritis with glomerular basal membrane disorder, proteinuria, and elevated serum creatinine levels in MRL/*lpr* mice. Collectively, the evidence strongly suggests that systemic hDPSC transplantation is a novel immunosuppressive approach for SLE treatment.

Although the mechanism of single systemic hDPSC transplantation has not been evaluated, some mechanisms explaining the therapeutic efficacy of singly transplanted hDPSCs in SLE-like MRL/*lpr* mice have been proposed [15,19,36]. The direct immunomodulatory action of hDPSCs on T lymphocytes may play a role in the therapeutic effects. Once donor hDPSCs are transplanted into recipient MRL/*lpr* mice, they must initially be targeted into the disease-specific tissues/organs. The local integration of transplanted hDPSCs then modulates the immune effects on neighboring recipient T lymphocytes at the disease site. However, only a negligible percentage of systemically transplanted hDPSCs are engrafted in the target tissues/organs [15,36]. Therefore, an in vivo immunomodulatory action through direct cell–cell contact seems far-fetched from the primary mechanism in single systemic transplantation of hDPSCs. However, the therapeutic action of singly transplanted hDPSCs seems to be mediated in recipient MRL/*lpr* mice via indirect communication through donor hDPSC-derived bioactive trophic factors and EVs rather than by direct communication with T lymphocytes. Recent studies have shown that hDPSCs can secrete various cytokines and chemokines with anti-inflammatory and anti-fibrogenic activities in the conditioned medium of hDPSCs (hDPSC-CM) [38]. Multiple administrations of hDPSC-CM containing multiple soluble factors have multiple effects on recipients with acute and chronic inflammation and autoimmune diseases [38]. However, the immunosuppressive mechanism of singly transplanted donor hDPSCs cannot be exploited by multiple administrations of hDPSC-CMs. These findings suggest that the paracrine effects of bioactive trophic factors released from donor hDPSCs do not contribute to long-lasting therapeutic effects in vivo. Another mechanism should be considered in the case of single systemic transplantation of hDPSCs in recipient MRL/*lpr* mice.

## 5. Telomerase Activity-Associated Niche Formation and Immunosuppressive Functions of Bone Marrow Mesenchymal Stem Cells (BMMSCs)

MSCs are often transplanted in situ together with various biomaterials to investigate the tissue-regenerative potential of mesodermal tissues, such as the bone. Human BMMSCs (hBMMSCs) are well-known clonogenic MSCs [60]. When hBMMSCs (4 × 10^6^ cells) were subcutaneously implanted with hydroxyapatite/beta-tricalcium phosphate (HA/TCP; 40 mg) as a carrier into immunocompromised mice [60], donor BMMSCs were found on the HA/TCP carrier and within the surrounding connective tissues. Some donor BMMSCs were differentiated into osteoblasts and directly deposited the de novo lamellar-bone-like matrix onto HA/TCP in the implant tissue. Interestingly, the connective tissue compartment surrounding the de novo bone-like matrix was gradually replaced by a bone marrow-like microenvironment associated with fewer hBMMSCs in the implant tissue. The bone marrow-like microenvironment consisted of two cell types: osteoblast-like cells lining the de novo bone-like matrix and leukocyte-like mononuclear cells. The lining of osteoblast-like cells initially originated from donor hBMMSCs, and, later, recipient-derived osteoblast-like cells appeared. Leukocyte-like cells were recruited only from recipient mononuclear cells. The mononuclear cells exhibited hematopoietic properties, such as hematopoietic colony formation and expression of hematopoietic cell markers, including Sca-1 and c-Kit stem cell markers and the CD45 lymphoid progenitor cell marker [61,62]. Mononuclear cells can recruit and systemically circulate in the recipient body [61,62], indicating that the de novo microenvironment is implicated in the hematopoietic niche, likely in the bone marrow. These findings indicate that BMMSCs potentially serve as bone marrow-like hematopoietic niche-organizing cells, as well as bone-forming cells. However, the detailed mechanisms underlying the BMMSC-mediated organization of the bone marrow have not been evaluated. Therefore, this dynamic integration of bone marrow-like hematopoietic tissue compartments post-BMMSC transplantation occurs at least in part through intercellular communication between donor BMMSCs and recipient hematopoietic cells. Thus, subcutaneous transplantation of BMMSCs is a reliable method that can be used to investigate BMMSC-mediated hematopoietic reconstruction and function [35,63]. Since bone marrow also functions as a primary lymphoid organ, niche-forming BMMSCs may also contribute to the regulation of lymphocyte production and differentiation.

Telomerase is one of the RNA-dependent DNA polymerases and consists of a catalytic subunit of telomerase and RNA components as a template for telomere repeat. Telomerase reverse transcriptase (TERT) is responsible for maintaining telomere length and promoting chromosomal stabilization of telomeres [64]. Telomerase activity contributes to diverse cellular functions, lifespan, and senescence. Most healthy somatic cells do not express telomerase, whereas stem/progenitor cells exhibit varied telomerase activity [64]. Both telomere length and telomere levels regulate the critical characteristics of stem cells, such as self-renewal, pluripotency, and multipotency. Overexpression of the human TERT gene (*hTERT*) can induce abundant de novo formation of a bone-like matrix and bone marrow-like niche after the hBMMSCs are subcutaneously implanted with HA/TCP carriers into immunocompromised mice [65]. Some studies have reported that pharmacological treatment contributes to the de novo formation of the bone-like matrix and bone marrow-like niche of hBMMSCs. Erythropoietin (EPO) is a glycoprotein hormone produced in the kidney that stimulates red blood cell production [66]. Recent studies demonstrate that EPO can enhance the bone formation of osteoblasts [67,68]. The recombinant EPO is a pharmacological indicator for secondary anemia in patients with chronic kidney disease and cancer. Recombinant EPO enhanced telomerase activity and bone marrow-like niche formation caused by BMMSCs in vivo [69]. Acetylsalicylic acid can also enhance telomerase activity and de novo bone-like matrix formation and bone marrow-like niche formation caused by BMMSCs [70]. Thus, the niche-forming function of recipient BMMSCs is regulated by telomerase.

BMMSCs from both patients with SLE and SLE model mice showed less stem cell characteristics of self-renewal and multipotency [20,71]. Disease-specific BMMSCs fail to exhibit telomerase activity and niche-forming capacity. SLE is caused by multiple factors, including genetic and environmental factors, and the complex factors induce abnormal lymphocytes to affect the functions of recipient BMMSCs. Thus, the niche-forming function of recipient BMMSCs is involved in the development and progression of SLE. The immunosuppressive function of BMMSCs is also regulated by telomerase. BMMSCs from *Tert*-knockout mice cannot inhibit T lymphocyte proliferation in vitro or in vivo [72]. MRL/*lpr* mouse-derived BMMSCs (*lpr*-BMMSCs) expressed reduced telomerase activity and suppressed Th17 cell differentiation [20]. Thus, targeting telomerase activity in recipient BMMSCs may be a novel therapeutic option for SLE treatment.

To understand hematopoietic niche formation and the immunomodulatory functions of recipient mouse BMMSCs from systemically hDPSC-transplanted MRL/*lpr* mice (hDPSC-*lpr*-BMMSCs), hDPSC-*lpr*-BMMSCs and *lpr*-BMMSCs were applied in the subcutaneous transplantation system with HA/TCP carriers into immunocompromised mice. Eight weeks after implantation, hDPSC-*lpr*-BMMSCs restored the formation of de novo bone marrow-like hematopoietic tissue components. hDPSC-*lpr*-BMMSCs further aided the accumulation of hematopoietic cells, including Sca-1-, c-Kit-, and CD45-positive cells in the implants, compared to the *lpr*-BMMSCs [35]. In comparison to *lpr*-BMMSCs, hDPSC-*lpr*-BMMSCs also improved both in vitro formation of hematopoietic colonies and in vitro immunomodulatory functions by decreasing Th17 cells, increasing Tregs, and enhancing CD4^+^AnnexinV^+^7AAD^+^ apoptotic T lymphocytes [35]. When hDPSC-*lpr*-BMMSCs and *lpr*-BMMSCs were systemically infused into MRL/*lpr* mice, hDPSC-*lpr*-BMMSCs ameliorated SLE-like disorders, including abnormal levels of serum autoantibodies, such as anti-dsDNA IgG, anti-dsDNA IgM, and ANA. hDPSC-*lpr*-BMMSCs further improved the renal conditions, including urine protein and serum creatinine levels and peripheral T lymphocyte levels, including Th17 cells, Tregs, and the Treg/Th17 ratio [35]. The systemic adoptive transfer of T lymphocytes (1 × 10^6^ cells per mouse) causes lethal immune reactions in immunocompromised mice [35]. hDPSC*-lpr*-BMMSCs systemically infused at 1 × 10^6^ cells per mouse into T lymphocyte-adopted immunocompromised mice prolonged their lifespan. However, these in vivo immunosuppressive functions were not observed after *lpr*-BMMSC transplantation into MRL/*lpr* and T lymphocyte-adopted immunocompromised mice [35]. Telomerase activity was suppressed in hDPSC-*lpr*-BMMSCs by functional knockdown using small interfering RNA (siRNA) for *Tert* (siRNATert) [35]. Pretreatment with siRNATert attenuated the in vivo hematopoietic niche-organizing ability, and in vitro and in vivo immunoregulatory functions of hDPSC-*lpr*-BMMSCs following their transplantation into immunocompromised, MRL/*lpr*, and T lymphocyte-adopted mice [35]. Thus, these results suggest that telomerase activity participates in the hematopoietic niche formation and immunomodulatory functions of recipient BMMSCs. Telomerase activity of recipient BMMSCs may be a novel target for SLE treatment.

## 6. Action of EVs Released from hDPSCs (hDPSC-EVs)

Most somatic cells release EVs, which are packed in the cell membrane of the lipid bilayer. EVs are divided into two groups based on their size: small exosomes (40–100 nm in diameter) originating from endosome compartments and large microvesicles (100–1000 nm in diameter) formed by budding from the parent cell membrane. Both types of EVs are involved in intercellular communication and signal transduction [73]. EVs released from the parent cells are transported to neighboring remote cells through bodily fluids, such as blood and tissue fluids. The transported EVs are used to deliver EV-containing contents to target cells. Thus, intercellular communication between parental and target cells is transmitted by the membrane fusion of EVs. The internal and/or membranous contents of EVs serve as intercellular communication factors and epigenetically regulate signal transduction or gene expression to control the cellular functions of target cells.

hDPSCs can also release hDPSC-EVs, including exosomes and microvesicles, into hDPSC-CM [35,62,74]. hDPSC-EVs are packed within a lipid bilayer membrane. The particle size of hDPSC-EVs ranges from approximately 50 to 500 nm in diameter. hDPSC-EVs highly express CD9, CD63, and CD81, but not CD90, on their vesicular membrane [35,62]. They contain various components, including proteins, such as cytokines and chemokines, and nucleic acids, such as mRNAs and non-coding RNAs, especially microRNAs. They also contribute to intercellular communication between donor and recipient cells under physiological and pathological conditions [35,62]. The released hDPSC-EVs can also fuse to the cell membrane of recipient cells and transfer their nucleic acid components to regulate the functions of recipient cells (Figure 2). Thus, hDPSC-EVs epigenetically regulate the transcription and translation of mRNAs and proteins in target cells. The mechanism of hDPSC-EV secretion is currently unknown. RAB27A, a small GTPase, plays a vital role in regulating vesicle trafficking to secrete EVs from the parent cells at inflammatory sites [66]. A recent functional knockdown study showed that RAB27A could regulate the secretion of hDPSC-Evs from parental hDPSCs [35].

## 7. Effects of Systemically Administrated hDPSC-Evs on the Telomerase Activity-Associated Niche Formation of Recipient *lpr*-BMMSCs in MRL/*lpr* Mice

The negligible frequency of engrafted hDPSCs in the bone marrow of MRL/*lpr* mice [35,62] suggests that indirect communication via hDPSC-Evs may be involved in the therapeutic rescue of BMMSC functions in these mice. To examine whether hDPSC-Evs may be involved in the therapeutic efficacy of hDPSCs, *RAB27A* siRNA (siRNARAB27A)-treated hDPSCs were systemically transplanted into MRL/*lpr* mice [75]. Four weeks after transplantation, the siRNARAB27A-hDPSCs attenuated the therapeutic efficacy of autoantibody hyperproduction, renal dysfunction, and the abnormal Treg/Th17 ratio in systemic hDPSC transplantation into MRL/*lpr* mice [35], suggesting that hDPSC-EVs are involved in the therapeutic efficacy of systemic hDPSC transplantation into MRL/*lpr* mice.

A recent study demonstrated the therapeutic effects of hDPSC-EVs in MRL/*lpr* mice [35]. hDPSC-EVs were systemically administered at 100 μg per mouse into MRL/*lpr* mice (female, 16-week-old mice). Four weeks after systemic infusion, the administration of hDPSC-EVs significantly recovered serum levels of autoantibodies, including anti-dsDNA and ANA antibodies, and improved renal disorders, including nephritis and proteinuria, and serum creatinine levels in MRL/*lpr* mice. hDPSC-EV administration significantly improved the systemic immune condition of Th17 cells, Tregs, and the Treg/Th17 ratio in MRL/*lpr* mice. The RNA contents were protected from RNase exposure and were stable within EVs, such as hDPSC-EVs. When hDPSC-EVs are treated with RNase, the RNA content is significantly diminished within the EVs with membrane structures [35]. Therefore, RNase pre-treatment may be of assistance in investigating the function of RNA contained within hDPSC-EVs. Once RNase-treated hDPSC-EVs (RNase-hDPSC-EVs) were systemically infused at 100 μg into MRL/*lpr* mice, the therapeutic efficacy of hDPSC-EVs in MRL/*lpr* mice was attenuated. In vivo cell tracking analysis revealed that fluorescent-labeled hDPSC-EVs were localized in recipient bone marrow cells seven days after administration. *TERT* expression and telomerase activity improved in recipient BMMSCs from hDPSC-EV-infused MRL/*lpr* mice (EV-*lpr*-BMMSCs) but not in recipient BMMSCs from RNase-hDPSC-EV-infused MRL/*lpr* mice (RNase-EV-*lpr*-BMMSCs). A subcutaneous transplant assay showed that the EV-*lpr*-BMMSC transplant group showed an expanded de novo bone marrow-like niche structure and increased Sca-1-, c-kit-, and CD45-positive cells. RNase-EV-*lpr*-BMMSCs attenuated EV-*lpr*-BMMSC enhancement.

## 8. Effects of Systemically Administered hDPSC-EVs on the Immunomodulatory Functions of Recipient *lpr*-BMMSCs

Recipient BMMSCs can modulate the inductive levels of Tregs and Th17 cells following co-culture with anti-CD3 and anti-CD28 antibody-activated mouse T lymphocytes [35]. *lpr*-BMMSCs increased the levels of Th17 cells and decreased the levels of Tregs and CD4^+^ apoptotic cells. EV-*lpr*-BMMSCs affected the in vitro levels of T lymphocytes. EV-*lpr*-BMMSCs reduced the levels of Th17 cells and enhanced the levels of Treg and CD4^+^ apoptotic cells. When T lymphocytes isolated from MRL/*lpr* mice were infused into immunocompromised mice, their lifespan was significantly reduced. The reduced lifespan was not improved by *lpr*-BMMSC transplantation into T lymphocyte-adapted immunocompromised mice. EV-*lpr*-BMMSC transplantation prolonged the lifespan of the T lymphocyte-adopted immunocompromised mice. EV-*lpr*-BMMSCs improved SLE-like disorders in MRL/*lpr* mice, including a reduction in the levels of peripheral autoantibodies, such as anti-dsDNA IgG, anti-dsDNA IgM, and ANA antibodies, and an improvement in renal disorders, including nephritis; increased levels of proteinuria and serum creatinine and the systemic immune status of Th17 cells, Tregs, and the Treg/Th17 ratio. RNase-pretreated EV-*lpr*-BMMSCs (RNase-EV-*lpr*-BMMSCs) reduced the in vitro levels of T lymphocytes, including Th17 cells, Tregs, and CD4^+^ apoptotic cells. Furthermore, RNase-EV-BMMSCs completely attenuated the in vivo effects of EV-*lpr*-BMMSCs in the SLE MRL/*lpr* mouse model and T lymphocyte-adopted immunocompromised mice. Thus, these findings suggest that the RNA content of hDPSC-EVs rescues the hematopoietic niche-forming and immunoregulatory functions of recipient BMMSCs via telomerase activity in MRL/*lpr* mice.

## 9. Mechanism of hDPSC Transplantation via Recovering Telomerase Activity of hDPSC-EVs in SLE

Single administration of hDPSC-EVs continuously recovered the mRNA levels of *Tert* in the recipient BMMSCs in ovariectomized (OVX) and MRL/*lpr* mice, resulting in improved BMMSC function by regulating telomerase activity. However, RNase-pretreated hDPSC-EVs attenuated the effects of hDPSC-EVs on the recipient BMMSCs [35,59]. The expression of TERT mRNA is regulated by epigenetic modifications, including histone acetylation and methylation, in its promoter region [76,77,78]. MIR346 binds to a region in the 3′UTR of TERT mRNA in human cervical cancer cells, leading to an upregulation of TERT expression [79]. Recently, we detected MIR346 in hDPSC-EVs and its parent hDPSCs [35]. We also demonstrated that hDPSC-EV treatment enhanced the expression of MIR346 in BMMSCs. RNase pretreatment attenuated the expression of MIR346 in hDPSC-EVs and the enhancement of MIR346 expression in hDPSC-EV-treated hBMMSCs [35]. Thus, hDPSC-EV-contained MIR346 might participate in epigenetically regulating TERT mRNA expression in recipient BMMSCs. RNA depletion attenuated the advantage of hDPSC-EV administration to the hematopoietic niche-forming and immunomodulatory functions of recipient BMMSCs via the rescuing of Tert mRNA expression and its associated telomerase activity in MRL/*lpr* mice [35], suggesting that the RNAs within hDPSC-EVs can epigenetically regulate the TERT/telomerase activity pathway to recover the functions of hematopoietic niche formation and immune regulation in recipient BMMSCs (Figure 2). Recent studies have suggested that hDPSC-EVs contain multiple small RNAs, including miRNAs, which act as an immunomodulator to exert immunosuppressive effects [73]. Thus, the interactive regulation of multiple RNAs within hDPSC-EVs could be proposed as a mechanism of single hDPSC transplantation in MRL/*lpr* mice. These findings suggest that hDPSC-EVs may play a crucial role in the biological crosstalk between donor hDPSCs and recipient BMMSCs to achieve a therapeutic mechanism (Figure 2 and Table 2).

## 10. Conclusions

Collectively, recent findings indicate that cell–cell communication via multiple RNA(s) within hDPSC-EVs released from hDPSCs is involved in the therapeutic mechanism of the systemic transplantation of hDPSCs in SLE MRL/*lpr* mouse models. The RNA(s) within hDPSC-EVs target recipient BMMSCs to epigenetically regulate the expression of the *Tert* gene. This enhances hematopoietic niche formation and rescues the immune microenvironment associated with revitalized Tert-associated telomerase activity. These findings provide novel insights into the mechanism of hDPSC-based therapy, revealing that intercellular communication between donor-derived cells and recipient cells plays a vital role in recovering the recipient pathological niche.

## Figures and Tables

**Figure 1 ijms-23-03479-f001:**
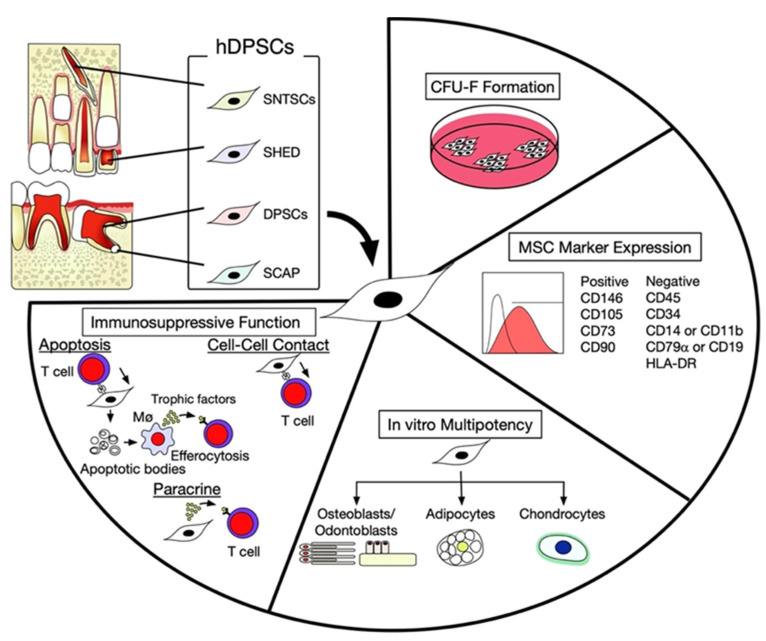
Class and characterization of human dental pulp-derived stem cells (hDPSCs). hDPSCs can be isolated from dental pulp tissues of deciduous, permanent, supernumerary teeth, and apical papillae of immature tooth roots, such as impacted wisdom teeth. hDPSCs exhibit mesenchymal stem cell (MSC)-like phenotypes and functions, including the adherent colony-forming capacity of colony-forming unit-fibroblasts (CFU-F), MSC marker expression, multipotent differentiation into osteoblasts, adipocytes, and chondrocytes, and immunosuppressive function. DPSCs, dental pulp stem cells; SCAP, stem cells from apical papilla; SHED, stem cells from exfoliated human deciduous teeth; SNTSCs, human supernumerary tooth-derived stem cells; Mø, macrophages.

**Figure 2 ijms-23-03479-f002:**
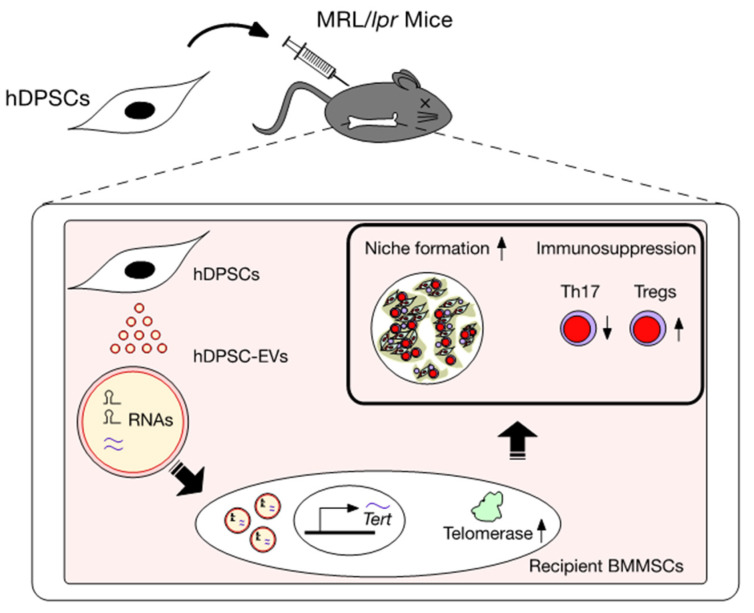
Immunosuppressive mechanism of hDPSC transplantation in systemic lupus erythematosus (SLE) model MRL/*lpr* mice. Systemically administrated donor hDPSCs release extracellular vesicles, hDPSC-derived Evs (hDPSC-Evs), and are incorporated into recipient bone marrow MSCs of MRL/*lpr* mice (*lpr*-BMMSCs). The RNA contained within the hDPSC-Evs are transferred into the *lpr*-BMMSCs to epigenetically enhance telomerase reverse transcriptase gene (*Tert*) expression, improving the SLE disorders of MRL/*lpr* mice via recipient BMMSC-mediated niche reconstruction and immunosuppressive function. Th17, interleukin 17 helper T lymphocytes; Treg, regulatory T lymphocytes.

**Table 1 ijms-23-03479-t001:** Molecules and extracellular vesicles (EVs) in immunosuppressive functions of hDPSCs.

**Cell-Cell Contact Molecules**	**References**
FASL/FAS	[14,15,19,36,39,40]
PD-L1/PD-1	[41,42]
**Paracrine factors and EVs**	**References**
HGF	[43]
HIF	[44]
IDO-1	[14,45,46,47]
IL-10	[48]
MCP-1	[49,50]
NO	[51]
PGE2	[14,39,40]
ROS	[51]
SIGLEC9	[49,50]
TGFB	[52]
EVs	[35,45,53,54,55]

HGF, hepatocyte growth factor; HIF, hypoxia-induced factor 1; IDO-1, indoleamine-2,3-dioxygenase 1; MCP-1, monocyte chemotactic protein 1; NO, nitric oxide; PGE2, prostaglandin E2; PD-1, programmed cell death protein 1; PD-L1, programmed cell death 1 ligand 1; ROS, reactive oxygen species; SIGLEC9, ectodomain of sialic acid-binding Ig-like lectin-9; TGFB, transforming growth factor beta.

**Table 2 ijms-23-03479-t002:** Key studies in hDPSC-based therapy for systemic lupus erythematosus (SLE).

Donor	Model	Interpretation	References
hDPSCs	In vitro co-culture Systemic TP into MRL/*lpr* mice	Th17 cell suppression, Treg induction Improvement in SLE-like disorders	[14,15,19,20]
hDPSCs	Systemic TP into OVX mice	FasL/Fas mediated T cell apoptosis	[36]
hDPSC-EVs	Systemic TP into OVX mice	Improvement in recipient BMMSC functions.	[63]
hDPSC-EVs	Systemic TP into MRL/*lpr* mice	Improvement in recipient mediated hematopoietic niche formation and hematopoiesis Improvement in SLE-like disorders	[35]

BMMSCs, bone marrow-derived mesenchymal stem cells; hDPSCs, human dental pulp-derived stem cells; hDPSC-EVs, hDPSC-released extracellular vesicles; OVX, ovariectomy; TP, transplantation.

## Data Availability

Data are available from the corresponding author on reasonable request.

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
