# Peer review of "A New Target of Dental Pulp-Derived Stem Cell-Based Therapy on Recipient Bone Marrow Niche in Systemic Lupus Erythematosus"

_ijms, 2022, doi:10.3390/ijms23073479_

Round 1
Reviewer 1 Report
-Title : Ok
- Abstract : Ok
- Keywords: Correct the keywords according to MeSH terms
- Introduction:
-The articles should also have a timeline for papers that have used for this review and
name the keywords and databases that have used for the search.
-Line 26: please more information’s on the autoimmune diseases
-Line 32: Where in Europe?
-The originality of the present paper?
-Characterization of hDPSCs
-Line 67: « Dentin is a unique hard tissue” ?? please revise
-Line 82: “hDPSCs contributed”
-Figure 1: is an original figure or a copied figure from another paper? If yes, the authors must ask for the authorization; Very small letters please revise
-Line 177: “multiple soluble factors”, I didn’t understand, please explain
-Line 184: I prefer to explain what is the BMMSCs in the title of this part and not only the abbreviation
-Line 215-222: need references
-Line 233: “Some studies”, and at the end of the sentence there is only one reference
-Line 263-287: need references
-Line 361: “modified from Sonoda et al.”, with authorization?
-Figure 3 legend: the same comment for Figure 2
References and text are well written according to MDPI style
Author Response
Reply to Comments and Suggestions from Reviewer 1
We are very thankful very valuable and concerned comments and suggestions to Reviewer 1. We have revised as described below.
Reviewer 1’s point 1
-Title : Ok
Reply to the point
Thank you for your positive comment.
Reviewer 1’s point 2
- Abstract : Ok
Reply to Comments and Suggestions 2
Reply to the point 2
Thank you for your positive comment.
Reviewer 1’s point 3
- Keywords: Correct the keywords according to MeSH terms
Reply to Comments and Suggestions 3
Reply to the point 3
Thank you very much for your suggestion. We have corrected the keywords on Lines 22 and 23 in the revised manuscript according to MeSH terms.
Reviewer 1’s point 4
- Introduction:
-The articles should also have a timeline for papers that have used for this review and name the keywords and databases that have used for the search.
Reply to the point 4
Thank you for your opinion. We have named the keywords and databases that have used for the search on Lines 26–28 in the revised manuscript according to the Reviewer 1’s suggestion. It is hard to have the timeline for papers because of many different and complicated articles.
Reviewer 1’s point 5
-Line 26: please more information’s on the autoimmune diseases
Reply to the point 5
Thank you for your point. We have added more information’s on the autoimmune diseases on Lines 29 and 30 in the revised manuscript.
Reviewer 1’s point 6
-Line 32: Where in Europe?
Reply to the point 6
Thank you for your comment. We have corrected the point to “Europa (France and Italy)” on Line 35 in the revised manuscript.
Reviewer 1’s point 7
-The originality of the present paper?
Reply to the point 7
Thank you so much for your deep concern. The originality of the present paper is an offer to he proposed mechanism of therapeutic effects of systemic transplantation of hDPSCs for SLE. Thus, we have revised the current sentence on Line 44 in the original manuscript to the revised one on Lines 57 and 58 in the revised manuscript.
Reviewer 1’s point 8
-Characterization of hDPSCs
-Line 67: « Dentin is a unique hard tissue” ?? please revise
Reply to the point 8
Thank you for your point. We have revised the part to Lines 75 and 76 in the revised manuscript.
Reviewer 1’s point 9
-Line 82: “hDPSCs contributed”
Reply to the point 9
Thank you for your comment. We have revised the part “hDPSCs contributed to the in vivo formation of dentin-pulp complex and/or bone-bone marrow compartment” to “hDPSCs had a in vivo capability to form dentin-pulp complex” on Lines 91 and 92 in the revised manuscript.
Reviewer 1’s point 10
-Figure 1: is an original figure or a copied figure from another paper? If yes, the authors must ask for the authorization; Very small letters please revise
Reply to the point 10
Thank you for your comment. The figure is original. The letter size has been corrected in the revised Figure 1.
Reviewer 1’s point 11
-Line 177: “multiple soluble factors”, I didn’t understand, please explain
Reply to the point 11
Thank you for your question. Current explanation was insufficient. We have corrected this part to “Multiple soluble factors in hDPSC-CM simultaneously affect to the in vivo immunosuppressive effects of hDPSCs” on Lines 262 and 263 in the revised manuscript.
Reviewer 1’s point 12
-Line 184: I prefer to explain what is the BMMSCs in the title of this part and not only the abbreviation
Reply to the point 12
Thank you for your suggestion. We have corrected the part according to the Reviewer 1’s suggestion as Lines 269 and 270 in the revised manuscript.
Reviewer 1’s point 13
-Line 215-222: need references
-Line 233: “Some studies”, and at the end of the sentence there is only one reference
-Line 263-287: need references
Reply to the point 13
Thank you for your suggestion. We have added a sentence and references in the revised manuscript on Lines 381-406 in the revised manuscript according to the Reviewer 1’s suggestion.
Reviewer 1’s point 14
-Line 361: “modified from Sonoda et al.”, with authorization?
-Figure 3 legend: the same comment for Figure 2
Reply to the point 14
Figures 2 and 3 have been replaced as original ones in the revised manuscript. Both sentences in the current legend pointed by the Reviewer 1 have been removed from the revised legend of manuscript.
Reviewer 1’s point 15
References and text are well written according to MDPI style
Reply to the point 15
Thank you for your positive comment.
Reviewer 2 Report
The title of the manuscript is focus on dental pulp derived MSCs, however the literature support is based on both dental pulp and bone marrow MSCs.
Point 2. Characteristics of hDPSCs.
The very first paragraph is well known fact about stem cells and need to be removed.
Point 3. Intracellular interaction between hDPSCs and T lymphocytes.
A table should be introduced highlighting different molecules and factors which are involved in inflammation.
Mechanistic approach is missing in the manuscript. Authors need to focus on the mechanism involved in the regulation of inflammation by stem cells.
Figure 3 is irrelevant with current manuscript which need to be replaced with figure highlighting inflammation and its treatment with stem cells.
Literature need to be up to date with more than 70 percent citation within last five years.
Author Response
Reply to Comments and Suggestions from Reviewer 2
We are very thankful very valuable and concerned comments and suggestions to Reviewer 2. We have revised as described below.
Reviewer 2’s point 1
The title of the manuscript is focus on dental pulp derived MSCs, however the literature support is based on both dental pulp and bone marrow MSCs.
Reply to the point
Thank you very much for your deep concern to the current title and literature. In this manuscript, we discussed the therapeutic efficacy of systemic transplantation of dental pulp derived MSCs (hDPSCs) on systemic lupus erythematosus (SLE). The major proposed mechanism is that extracellular vesicles released form donor dental pulp derived MSCs target recipient bone marrow MSCs (BMMSCs) to improve the bone marrow niche-organizing function. Therefore, in this manuscript, we need much detail regarding to the functions of BMMSCs as therapeutic target of hDPSC transplantation to support the literature.
Reviewer 2’s point 2
Characteristics of hDPSCs.
The very first paragraph is well known fact about stem cells and need to be removed.
Reply to the point 2
Thank you so much for your suggestion. We have removed the first two sentences of first paragraph from the revised manuscript. The other sentences is necessary to explain mesenchymal stem cells
Reviewer 2’s point 3
Intracellular interaction between hDPSCs and T lymphocytes.
A table should be introduced highlighting different molecules and factors which are involved in inflammation.
Reply to the point 3
Thank you for your thoughtful opinion. We have introduced the issue and summarized as Table 1 in the revised manuscript.
Reviewer 2’s point 4
Mechanistic approach is missing in the manuscript. Authors need to focus on the mechanism involved in the regulation of inflammation by stem cells.
Reply to the point 4
We applicate the Reviewer 2 to her/his helpful suggestion. Since the explanation to mechanistic approach was insufficient in the current manuscript as pointed by the Reviewer 2, we have added a new section “9. Mechanism of hDPSC transplantation via recoverin telomerase activity of recipient BMMSCs by hDPSC-EVs in SLE” to explain the mechanism involved in the regulation of inflammation by hDPSCs from Line 523 to Line 561 in the revised manuscript.
Reviewer 2’s point 5
Figure 3 is irrelevant with current manuscript which need to be replaced with figure highlighting inflammation and its treatment with stem cells.
Reply to the point 5
Thank you for your valuable suggestion. We have replaced the current Figure 3 to the revised Figure 3 in the revised manuscript according to the reviewer 2’s suggestion.
Reviewer 2’s point 6
Literature need to be up to date with more than 70 percent citation within last five years.
Reply to the point 6
Thank you for your suggestion. We have rearranged the references in the revised manuscript as possible as we can according to the Reviewer 2’s suggestion.
Round 2
Reviewer 2 Report
The revised version is improved. The the queries were addressed by the authors. Therefore, I recommend this manuscript to be accepted.
Author Response
We are grateful for Reviewer 2's positive comments.